# Bladder Dysfunction and Re-Absorbable Bulking Agent Affect Success Rate in Children Underwent Endoscopic Treatment for Vesicoureteral Reflux: A Long-Term Follow-Up Study

**DOI:** 10.3390/children8100875

**Published:** 2021-10-01

**Authors:** Raffaella Cocomazzi, Alessia Salatto, Vittoria Campanella, Valentina Pastore, Cosetta Maggipinto, Gabriella Aceto, Fabio Bartoli

**Affiliations:** 1Pediatric Surgery Unit, Department of Clinic and Surgical Science, University of Foggia, 71122 Foggia, Italy; raffaellacocomazzi@alice.it (R.C.); alessia.salatto@unifg.it (A.S.); vittoriacampanella@libero.it (V.C.); cosettamaggipinto@libero.it (C.M.); 2Institute of Radiology, University of Foggia, 71122 Foggia, Italy; pastorevalentina@gmail.com; 3Pediatric Nephrology Unit, Children Hospital “Giovanni XXIII”, Policlinico of Bari, 70124 Bari, Italy; acetogabriella@alice.it

**Keywords:** vesicoureteric reflux, endoscopic treatment, children

## Abstract

This paper is designed to evaluate the results (at long-term follow-up of) children affected by dilating VUR. Our attention was focused on how VUR grade, laterality, bladder dysfunction (BD), the double renal system, and the type of bulking substance may affect VUR resolution in the long-term period. The charts of 93 children with dilating VUR who underwent endoscopic treatment (ET) and with a minimum post-operative follow-up of 7 years were reviewed (mean follow-up time was 9.6 + 1.4). The majority of patients had severe and bilateral VUR. Polydimetilsiloxane or hyaluronic acid/dextranomer (PDS or Ha/Dx) were used as bulking agents. VUR persistence following endoscopic injection was independent with respect to grade, laterality, duplex renal system, and BD. However, the rate of VUR persistence was significantly higher in children with BD. Children treated with Ha/Dx had a higher rate of VUR persistence. This research demonstrated that ET of VUR is also effective at very long term follow up (and without the development of significant complications). We also showed that patients treated with absorbable bulking agents such as Ha/Dx may experience a higher recurrence rate at the long-term follow-up). We also confirm that the only preoperative condition affecting VUR recurrence was bladder dysfunction.

## 1. Introduction

Vesicoureteral reflux (VUR) is the most common uropathy in children, affecting 1 to 3% of the general pediatric population [1]. The natural history of VUR is often poorly predictable despite several risk factors for persistence that have been suggested, such as VUR grade, age at diagnosis, coexisting bladder dysfunction (BD), recurrent UTIs, and the double renal system (DS).

In a recent and excellent review, Lackgren and Stenberg have analyzed the most relevant literature on different treatment options for VUR, concluding that endoscopic treatment (ET) had become the preferred treatment option for dilating reflux by parents and clinicians [2]. Furthermore, those authors stated that a multifactorial assessment of VUR is needed to improved patient selection and outcome. Nowadays, the major debates about the ET of VUR are focused on two aspects: the best bulking agent and the results at long-term follow-up.

This research aimed to evaluate the results at long-term follow-up of children affected by dilating VUR who underwent endoscopic injection over 12 years. Our attention focused on how VUR grade, laterality, BD, DS, and type of bulking substance (PDS or Ha/Dx) may affect VUR resolution in the long term. 

## 2. Materials and Methods

We reviewed only the charts of 93 children with dilating VUR who underwent ET and with a minimum post-operative follow-up of 7 years (mean follow-up time was 9.6 ± 1.4). The follow-up period started after the last required in-hospital exam. Forty-seven were male, while 46 were female. The mean age at the time of first endoscopic injection was 4.5 ± 2.8 years. All children that, at diagnosis, had only non-dilating VUR were excluded from the study. Furthermore, patients with neurogenic BD secondary to myelomeningocele or other primary neurological diseases were excluded. The most indications for ET were severe VUR alone 37%, recurrent UTIs in 41% and miscellaneous (Reflux nephropathy, DS and persistent VUR) in 12% of cases. All children underwent a pre- and post-operative evaluation with blood and urine analysis, urine culture, renal ultrasound, micturing cystography (MCU), and a ^99Tc^ DMSA renal scan. On the postoperative period, all patients were kept on antibiotic prophylaxis for three months or until VUR had disappeared. At follow-up, all had monthly urine culture, renal ultrasound at one week, three months and one year. MCU was often replaced by a cystosonogram to minimize the risk of radiation and it was scheduled at three months and one year follow-up after each endoscopic procedure then every three years after VUR resolution (nowadays we are not planning this further control). BD was defined as abnormalities in either filling/emptying of the bladder, requiring both treatment and diagnostic follow-up (in this study we have not analyzed data according to the specific type of BD). History of recurrent UTIs was recorded in 58/93 patients (62%) before ET. We use the definition of febrile UTI as reported in the randomized intervention for children with vesicoureteral reflux study [3]. Unfortunately, long term follow-up evaluation of recurrent UTIs and renal function progression was not included in this study since several patients were lost to follow-up for several reasons such as parents moving to other cities and non-compliance with medical examinations/data recording.

According to the International Grading System Study Group for Vesicoureteral Reflux, we have considered as moderate (MOD) reflux those renal units with grade 3 VUR while as severe (SEV) those with grade 4 and 5. Grade 3 to 5 were also defined as dilating reflux.

### 2.1. Informed Consent

For all patients was obtained informed consent was obtained from parents regarding the type of treatment proposed, postoperative treatment, follow-up laboratory and instrumental tests. Furthermore, we obtained consent to use sensible private data for scientific reports or communications to meetings. This retrospective study was waived for approval by the Ethical Commission since it was based only on data collected from clinical charts with permission to use private information for scientific purposes. Any of the patients underwent additional tests for the purpose of study. 

### 2.2. Endoscopic Injection Procedure 

A single operator performed all endoscopic procedures under general anesthesia with an 8 Fr/10 Fr Pediatric Cystoscope between 1996 and 2010. The type of bulking agent was addressed mainly on the basis of parents’ choice for absorbable and nonabsorbable substances. Further, during the initial experience only non-absorbable substance were available. The injection of the bulking agent was guided by the insertion of a ureteral catheter into the ureteral lumen. When the urethral meatus was too large, we chose the hydrodistension technique to achieve a better result. No technical changes were made over the years regarding the endoscopic injection technique. Based on our protocol, ET was offered three times before considering it as failed for both bulking agents. Children who failed Ha/Dx underwent an additional single attempt with PDS as a rescue therapy. If this single treatment also failed, open ureteral re-implantation would have been the only option.

Long-term follow-up was mainly based on regular ultrasound and clinical evaluation. VUR resolution was demonstrated in long-term follow-up by cystosonogram. Macrohematuria, unexplained high fever, febrile UTI, and hydronephrosis were all considered indications for patient re-evaluation. Positive urine culture was defined when the colonies/mL value was 10^5^ or greater or there was any amount of pseudomonas or klebsiella. Urine collection was by catheterization or middle miction urine sample in older children. 

The statistical analysis was performed with SPSS Statistics 18.0 (SPSS Inc., IBM Company, Chicago, IL, USA) for paired t-test and Chi-square test. A *p*-value of less than 0.05 was considered statistically significant.

## 3. Results

Unfortunately, after seven years of follow-up, only 50% of the children included in this study had a ^99tc^ DMSA renal scan at the time of the last follow-up and 60% stopped to fill the UTI diary after three years from proved VUR resolution. However, all patients included in the study had documented VUR status by MCU or cystosonogram. On the post-operative follow-up, we classified the results for VUR control in A) Resolution (no VUR) and B) Persistence (persistence of VUR at any grade).

Table 1 reports the main clinical characteristics of VUR patients included in the study according to patients number, VUR grade, bilateral (BIL) or unilateral (UNIL), BD, reflux nephropathy (RN = split renal function < 45%), and coexisting DS. The bilateral forms were prevalent (BIL vs. UNIL; 56% vs. 44%, *p* < 0.05) and the renal units (RU) with severe VUR were more represented in the unilateral cases (UNIL: SEV 66% vs. MOD 34%, *p* < 0.01). Overall, RU with severe VUR was more numerous than that with moderate VUR (RU: SEV 56% vs. MOD 44%, *p* < 0.05).

Table 2 gives the distribution of patients with coexisting VUR and BD/RN/DS based on laterality and severity of VUR. We notice that severe and bilateral VUR were more common in patients with BD (BD: SEV 48% vs. MOD 30%, *p* < 0.05; BD: BIL 59% vs. UNIL 41%, *p* < 0.05), while no significant difference was found when we compared with the number of patients with BD/the total number of children based on laterality and severity of VUR. However, when analyzing the same parameters on RN, we failed to find significant statistical differences between groups. In the group of patients with DS, the only statistically significant difference was found between DS with severe VUR vs. moderate VUR (DS: SEV 53% vs. MOD 20%, *p* < 0.05). All cases of DS were unilateral.

After ET, the overall VUR persistence rate was observed in 24/145 RU (17%) and in 20/93 (22%) patients. Table 3 reports the incidence of persistence rate in several subgroups. We were unable to demonstrate statistical significance except for patients with BD who had an increased VUR persistence compared with the overall success rate (VUR Persistence rate: TOT 22% vs. BD 28%, *p* < 0.05). Despite a higher percentage of persistent VUR in DS patients when compared with those of all patients, it could not reach a statistically significant difference because of the limited number of patients with associated DS. Transitory ureteral obstruction following endoscopic injection was observed in 7 RU/145 RU (5%), requiring double J stent placement for 8–12 weeks. After stent removal, all patients had progressive improvement of ureteral dilatation. 

### 3.1. Results According to Bulking Agent: Polydimethylsiloxane or Hyaluronic Acid/Dextranomer

Of the 93 patients (pts), 63 pts/101 RU and 30 pts/44 RU have been treated with PDS and Ha/Dx, respectively. There were not significant differences between the two groups in term of percentage of children with severe VUR, associated BD and DS.

### 3.2. Results with Ha/Dx

The incidence of persistent VUR was observed in 13/30 pts (43%) and 20/44 RU (45%). Seven of them underwent rescue therapy with PDS. Six of seven were successfully treated, while one underwent ureteral re-implantation.

### 3.3. Results with PDS

Sixty-three pts/101 RU received PDS as the primary treatment, while an additional seven pts/12 RU as rescue therapy following failed Ha/Dx injection. Sixteen out of sixty-three pts had persistent VUR (25%) and 24/101 RU (24%). The incidence of persistence in the Ha/Dx group was significantly higher than that with the PDS group (VUR persistent rate: PDS 25% vs. Ha/Dx 43%, *p* < 0.05). Similar results were observed when the persistence rate was calculated on the number of RU. 

Five children required a technically demanding ureteral re-implantation that was successfully performed in all of them. 

## 4. Discussion

First, we are conscious of some weak points of this study, mainly due to some patients lost to follow-up or not well recorded by a local nephrologist. Furthermore, it has the usual limitations of a retrospective study relative to a prospective study design. Endoscopic treatment of VUR, since its initial report in 1981 by Matouschek [4] and popularization by O’Donnel and Puri [5], has been investigated with respect to several types of research trying to better understand its efficacy and relevance of the bulking agent used. Initially, the not absorbable substance PTFE (TEFLON^®^) was the most popularized agent, but progressively, it was abandoned due to the risk of distant migration. As an alternative, polydimethylsiloxane (Macroplastique^®^) gained popularity as a nonabsorbable substance since it had a lower risk of migration. This characteristic was a consequence of the bigger particles that could not be fagocytated by macrophages [6,7]. In a previous manuscript, we reported our experience in treating any grade of VUR with PDS as a bulking agent with a nearly 90% success rate [8]. However, the concern for using permanent bulking agents has stimulated the diffusion of absorbable substances, of which the most widespread is dextrane copolymer/Hyaluronic acid. The main characteristics of Ha/Dx are biocompatibility, not immunogenic, not cancerogenic, and not migrating. In the last 20 years, several authors have reported different results with Ha/Dx mainly due to different injection techniques and experiences [9], VUR grade [10], young age [11], bladder function [12], and length of follow-up period [9]. Recently, Chertin et al. reported a success rate in the treatment of VUR ranging from 68% to 92% [13]. However, Blais et al. have reported a decreased efficacy of Ha/Dx over time due to its decrease in volume [9]. However, recently, a success rate of 85% has been reported by Harper et al. among children who underwent endoscopic injection of Ha/Dx with a follow-up period longer than 10 years [14]. Several authors have compared the efficacy of these two bulking agents. In 2002, Oswald et al. reported a similar success rate after a single injection of PDS and Ha/Dx, being 86.2% and 71.4%, respectively [15]. After three years of follow-up, Stredele et al. have reported VUR recurrence rates of 45.5% and 21.5% with PDS with Ha/Dx, respectively [16]. Bae et al. did not confirm these findings but underlined that in severe VUR, PDS was more effective [17]. Recently, Moore and Bolduc, in a study on long-term follow-up (mean 4.3 years), showed slightly better results in terms of VUR resolution with PDS (90%) vs. Ha/Dx (81%) [18]. 

Furthermore, Fuentes et al., evaluated the factors affecting the recurrence rate after three years of follow-up. They included the use of Ha/Dx as bulking as a variable associated with VUR recurrence together with high-grade reflux, treatment at an early age and BD [19]. Leung et al. have recently reported, after 60 months of follow-up, a resolution rate following Ha/Dx injection, which was differentiated according to VUR grade (63% III, 40% IV and 70% V) [20]. However, it is still unclear which substance (PDS or Ha/Dx) provides better long-term results.

Despite the large volume of literature dealing with the ET of VUR, there are only a few manuscripts considering the results over five years of follow-up. Our study seems to be one with the most long-term follow-up period comparing not-absorbable vs. re-absorbable bulking agents. Furthermore, it considers only grade moderate to severe VUR and two different bulking agents. Our population of patients was characterized by a larger number of children with severe and bilateral VUR; nearly a quarter of them were in treatment for BD. The first interesting data resulting from our study is that, following ET, successful long-term results are very common, especially in the group of patients treated with PDS. On the contrary, the percentage of persistence VUR in those children treated with Ha/Dx was significantly higher. Another important finding was that the success rate was independent of VUR severity and bilateralism and the association with DS. This confirms our previous initial studies, where all these reflux grades, laterality [8] and coexistence with duplex ureter [21] were not considered a risk factor for VUR persistence.

In our study, the only preoperative condition affecting the recurrence rate was BD. Furthermore, children with associated BD were the only patients who needed ureteral re-implantation. This finding was different from our previous report [8], where we did not show differences in the success rate between patients with or without BD. We believe that the shorter follow-up (about two years) in the previous study contributed to these different results. Authors have reported that severe form of BD carries the high risk of VUR recurrence following surgical treatment [22,23]. In a more recent study on ET of VUR, milder forms of voiding LUT dysfunction did not influence the results of ET for VUR [24], in which the dysfunction disappeared after cessation of the reflux. The authors suggest that the reflux was an underlying cause of the dysfunction in these cases. Other authors reported that the success rate was lower after a second injection in children with BD [25].

In conclusion, this research showed that ET of VUR is also effective in the very long term to follow up without the development of significant complications. We also observed that patients treated with absorbable bulking agents, such as Ha/Dx, might experience a higher recurrence rate in the long-term follow-up. In these patients, rescue therapy with PDS or ureteral re-implantation is the only viable alternative. We also confirm that reflux grade, bilateralism of VUR, or coexistence of duplex renal system should not be of concern for the future outcome. On the contrary, BD should be considered a risk factor for VUR recurrence. Finally, in our opinion, endoscopic injection for the treatment of VUR remains the first surgical choice in those children since it is minimally invasive, safe, and effective.

## Figures and Tables

**Table 1 children-08-00875-t001:** Clinical characteristics of children with VUR.

	Pts	VUR MOD	VUR SEV	VUR BIL	VUR UNIL
Total number of pts	93	-	-	-	-
RU with VUR	-	65	80	-	-
VUR BIL	52	18	20	-	-
VUR UNIL	41	15	26	-	-
VUR + BD	25	7	14	16	9
VUR + RN	16	7	7	9	7
VUR + DS	12	3	9	4	8

Here are reported the main clinical characteristics of vesicoureteral (VUR) patients included in the study according to patients number, VUR grade, bilateral (BIL) or unilateral (UNIL), bladder dysfunction (BD), reflux nephropathy (RN) and double renal system (DS). Only significant *p* values have been reported in results paragraph.

**Table 2 children-08-00875-t002:** Distribution of VUR according to grade and laterality in different groups of patients with RN, DS and BD.

	Pts with RN/Total Pts 16/93 (17%)	Pts with DS/Total Pts 12/93 (13%)	Pts BD/Total Pts 25/93 (27%)
VUR MOD	7/33 (22%)	3/32 (9%)	7/33 (22%)
VUR SEV	7/46 (15%)	9/46 (19%)	14/46 (30%)
VUR BIL	9/52 (17%)	4/52 (8%)	16/52 (31%)
VUR UNIL	7/41 (17%)	8/41 (20%)	9/41 (22%)

Here are reported the rate of reflux nephropathy (RN), duplex system (DS) and bladder dysfunction (BD) related to the total number of patients, the distribution according to vesicoureteral (VUR) grade (moderate = MOD or severe = SEV) and laterality (bilateral = BIL or unilateral = UNIL). Only significant *p* values have been reported in the results paragraph.

**Table 3 children-08-00875-t003:** Persistence rate at long term follow up in relationship to number of renal unit (RU) and patients (PtS).

	Persistance VUR/RU	Persistance VUR/PTS
Total	17%	22%
VUR MOD	18%	-
VUR SEV	15%	-
VUR UNIL	-	20%
VUR BIL	-	23%
Bladder Dysfunction	-	28%
Duplex Renal System	-	33%
Ha/Dx	45%	43%
PDS	24%	25%

Here is reported the distribution of persistence rate according to VUR grade (MOD/SEV), Laterality (BIL/UNIL), Bladder dysfunction, duplex renal system and bulking agent (Ha/Dx and PDS). Only significant *p* value have been reported in results paragraph.

## Data Availability

Not applicable.

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
