# Peer review of "Bladder Dysfunction and Re-Absorbable Bulking Agent Affect Success Rate in Children Underwent Endoscopic Treatment for Vesicoureteral Reflux: A Long-Term Follow-Up Study"

_children, 2021, doi:10.3390/children8100875_

Round 1

Reviewer 1 Report

This is an interesting report, where authors highlight the importance of long-term follow up in children underwent endoscopic treatment for VUR.

The introduction is clear and the method description is overall informative. I would describe statistical analysis in the method section and not in the results.

The results are not completely clear, particularly I would modify the tables in order to facilitate the understanding, maybe unifying in only one table that summarize the main results. 

In the discussion I would synthesize the first paragraph in which authors reviewed the literature and expand the comments on their original results, particularly bladder dysfunction.

Furthermore, there are several grammatical errors that must be corrected and English syntax  should be improved.

Author Response

1. I would describe statistical analysis in the method section and not in the results= Done, SA has been moved from results to Methods.

2. Unifying in only one table that summarize the main results= Done, table 2,3,4 have been unify in actual table 2

3. expand the comments on the original results, particularly bladder dysfunction= Done, I have expanded the discussion in the part about effects of bladder dysfunction on VUR resolution after endoscopic procedure

4. There are several grammatical errors and English synthax should be improved= Done, the manuscript has been submitted for English revision to American Editors Manuscript

Reviewer 2 Report

This is a retrospective review trying to answer a interesting question about the long term follow up after cystoscopic anti reflux surgery. Unfortunately this report lacks a lot of necessary information. Demographics of the patients, postoperativ management, ethical approval just to mention a few.

l recommend to rewrite this paper after getting information on how to do a retrospective study 

Author Response

Lack of demographics of patients, postoperative management and ethical approval= Done,  I have expanded all the information required and underlined the position of our Ethics Commission about waive for approval for retrospective study which did not required additional test as long as the consent was already obtained for publication of sensitive data for research or meetings(SEE INFORMED CONSENT PARAGRAPH).

Basically, quite most of the manuscript has undergone substantial rewriting